# An Improved Model for the Evaluation of Groundwater Recharge Based on the Concept of Conservative Use Potential: A Study in the River Pandeiros Watershed, Minas Gerais, Brazil

**Marcelo Alvares Tenenwurcel [1], Maíse Soares de Moura [1], Adriana Monteiro da Costa [1] , Paula Karen Mota [1] , João Hebert Moreira Viana [2] , Luís Filipe Sanches Fernandes [3] and Fernando António Leal Pacheco [4],***

[1]   Federal University of Minas Gerais, 6620 Antônio Carlos Ave., Pampulha, Belo Horizonte, MG 31270-901, Brazil; m-alvares@hotmail.com (M.A.T.); maisedemoura2013@gmail.com (M.S.d.M.); drimonteiroc@yahoo.com.br (A.M.d.C.); paulakarenmota@gmail.com (P.K.M.)

[2]   Brazilian Agricultural Research Corporation (Embrapa Maize and Sorghum), Sete Lagoas, MG 35701-97, Brazil; joao.herbert@embrapa.br

[3]   CITAB—Centre for the Research and Technology of Agro-Environment and Biological Science, University of Trás-os-Montes and Alto Douro, Quinta de Prados Ap. 1013, 5001-801 Vila Real, Portugal; lfilipe@utad.pt

[4]   CQVR—Chemistry Research Centre, University of Trás-os-Montes and Alto Douro, Quinta de Prados Ap. 1013, 5001-801 Vila Real, Portugal

*   Correspondence: fpacheco@utad.pt

**Abstract:** Water resources have been increasingly impacted due to the growth of water demand associated with environmental degradation. In this context, the mapping of groundwater recharge potential has become attractive to water managers as it can be used to direct public policies and conserve this natural asset. The present study modifies (improves) a spatially explicit model to determine groundwater recharge potential at the catchment scale, testing it in the Pandeiros River basin located in the state of Minas Gerais, Brazil. The model is generally based on the water balance approach and the input variables were compiled from institutional sources and processed in a Geographic Information System. The novelty brought by the aforementioned modification relates to the coupling of physical variables (conventional way) and land management practices (introduced here) in the estimation of a percolation factor. The role of land management practices for percolation was assessed by the so-called Conservative Use Potential (PUC) method, which classifies the areas of a river basin in terms of their potential for sustainable use. The results were validated by an independent method, namely the recession curve method based on the interpretation of hydrographs. In general, the groundwater recharge potential is favored in flat to gently undulating areas and forested regions, as well as where the landscape is characterized by well-structured soils, good drainage conditions and large hydraulic conductivity. The map of groundwater recharge potential produced in this study can be used by planners and decision makers in the Pandeiros River basin as a tool to achieve sustainable use of groundwater resources and the protection of recharge areas.

**Keywords:** groundwater recharge; water resources management; Conservative Use Potential; river basin; geographic information system; water balance

## 1. Introduction

Clean water resources are becoming scarcer due to the increasing demand for its use and to environmental degradation [1]. It has been estimated that 1/3 of all countries will have to adapt

their productive processes by 2025 because water is lacking, and more than 3 billion people might be living in regions under chronic draughts or water stress [2–6]. Owing to its great importance for society, economy and the environment, it is necessary to properly manage water resources. The increasing degradation of surface water resources not only in Brazil but all over the world [7], makes the sustainable management of groundwater become an even more essential activity. In this context, a precise evaluation of groundwater distribution entails the understanding of aquifer refilling processes and the quantification of groundwater inputs. This quantification is called groundwater recharge estimation. In light of this challenge, studies and maps related to the provision of water resources are needed, with the purpose of indicating priority areas for conservation or restoration and to direct public policies to protect this natural resource. In Brazil, groundwater recharge studies that exist are predominantly focused on the river basin scale [8].

The incorrect management of river basins can generate serious threats to water availability, hindering surface and groundwater, because the dynamic of hydrologic systems is vulnerable to human actions [9,10]. An example is the reduced capacity for water infiltration into the soil observed in areas that are occupied with civil constructions or impermeable pavements [10,11]. Other problems, such as the intensification of erosion or floods, can occur in the development of soil permeability declines. In rural catchments, the inadequate uses of the land, among other factors, can amplify the risk of flood occurrence [12] reducing the levels of recharge.

Changes in the dynamics of hydrologic systems are particularly worrying in aquifer recharge zones, which are regions that enable water infiltration and percolation toward an aquifer system, which is defined as the geological system capable of storing and distributing a significant amount of water [13–15]. Moreover, recharge zones can be defined as areas where the soil surface favors the water infiltration and percolation [16,17]. Water can also be retained in the soil and slowly recharge the aquifer [13,18,19].

Some recharge zones are more efficient than others and for that reason are called preferential groundwater recharge zones [20]. The environmental protection of these special areas is important to conserve the quality and quantity of water resources. Thus, detailed information on the groundwater recharge process can aid better land use and cover distribution, and indicate the best areas for agricultural activities with the lowest groundwater contamination risks caused by the release of substances with high polluting potential, such as pesticides.

Despite the importance of sustainable management of aquifer recharge zones [7,21–23], in the state of Minas Gerais, Brazil, this topic has not yet been properly studied. Therefore, a better comprehension about the factors that affect groundwater recharge is necessary, as well as the mapping of recharge areas that consider the sustainable management potential of the basin. These evaluations should be robust and contain physical–environmental factors [10], such as soil characteristics, geology, vegetation cover, climate, and topography. When all these data are evaluated together, they allow for sustainable water use, meaning a use without compromising groundwater recharge [10]. Besides, under these circumstances the volume of water withdrawn from the aquifer system can be defined according to its natural capacity [24].

Numerous groundwater recharge estimation methods exist. However, they all entail some level of uncertainty [25]. In general, the practical and conceptual limitations of recharge estimation models occur because the available hydrological and hydrogeological data are sparse or fragmented, and because the spatial and temporal variations in recharge are significant [26]. This difficulty is significant for semi-arid areas [27], causing recharge estimations in these areas to be even more challenging. There are direct and indirect methods to evaluate the groundwater recharge potential. The direct methods include geological and geophysical explorations, gravimetrical and magnetic models, and perforation tests [10]. The indirect methods include hydrological and hydrogeological models [28,29], using geographical information systems (GIS) combined with field work [30,31]. Other studies have employed different methods to estimate groundwater recharge, which are comprised of tracer methods, water table fluctuation models, lysimeter methods and simple water balance techniques. Some of

these studies have used numerical groundwater models or dynamically linked them to hydrological models to estimate recharge variations under different climate and land cover conditions [32–36]. For example, Döll (2008) modeled global groundwater recharge using the WaterGAP Global Hydrological Model (WGHM), which has failed to reliably estimate recharge in semi-arid regions [37]. In that study, the influence of vegetation was not taken into account, even though many studies have showed the importance of this variable for estimating the groundwater recharge [32,38–42]. Moreover, Chowdhury et al. (2010) delineated groundwater recharge zones in West Medinipur district, India, using a GIS approach mixed with remote sensing and multi-criteria decision making techniques [22]. The input variables considered in that study were geomorphology, geology, drainage density, slope and aquifer transmissivity. In general, the choice of a method should consider the precision level needed, the project execution viability, and the available financial resources.

Among methods available in the literature, Costa et al. (2019) proposed one for the evaluation of groundwater recharge potential based on the water balance approach that considers climatic variables, water runoff, and the percolation of water into the soil profile [10]. The authors obtained results for mean annual recharge similar to those calculated by the hydrograph recession curve analysis, which has been used as a validation method. Furthermore, they identified areas with larger recharge potential and suggested management practices to improve groundwater recharge in those areas, making their study a valuable tool for the sustainable use of groundwater and protection of recharge areas [10].

Despite the positive results obtained by Costa et al. (2019), it is worth noting that the land management practices were a consequence of groundwater recharge assessments, not a contributing factor to groundwater recharge included in the model. Indeed, all parameters included in Costa's water balance model were physical, while land management practices had no role, regardless of their potential to dynamically affect groundwater recharge [10]. Thus, the coupling of physical factors and land management practices in a recharge estimation model could be a motivation (and a novelty) for a subsequent study. Before the publication of Costa et al. (2019), Costa et al. (2017) [43] conducted a study in Minas Gerais and developed a method based on multi criteria analysis, which was efficient to map a so-called Conservative Use Potential (PUC). The PUC method weights a considerably large number of variables considering their importance for sustainable land use, including several variables linked to land management practices, such as drainage, soil depth and fertility, erosion potential, and land capability. Hence, one possible route to realize our research motivation would encompass including the PUC, as determined by Costa et al. (2017) [43], within the framework of the Costa et al. (2019) groundwater recharge method [10].

The general purpose of this study is therefore to take that step forward and embed the concept of PUC in the groundwater recharge method of Costa et al. (2019) [10]. In that method, a parameter is defined to measure water percolation based on the soil's effective porosity and hydraulic conductivity. The method presented in this work replaces porosity by three parameters strongly influenced by management practices, which are soil texture, drainage, and profile depth. The replacement has the specific purpose to check whether this set of variables responds more effectively to land use changes than the original variable (porosity). The model was tested in the Pandeiros River Basin (PRB), Brazil, to generate a spatially explicit map of groundwater recharge potential, at regional scale (1:100.000). This map has the potential to subsidize indications of preferred areas for restoration, recovery, and protection, ensuring a more sustainable water resources management in this basin.

## 2. Materials and Methods

### 2.1. Study Area

The Pandeiros River basin (PRB) is located in the northern state of Minas Gerais, Brazil, and has an area of 396,028 ha, which encompasses part of Januária, Bonito de Minas, and Cônego Marinho municipalities (Figure 1). The climate of the region is predominantly dry with mean annual temperature of 24.6 °C, which occasionally reaches a maximum of 33 °C in October, and a minimum of 14 °C in

July [44]. Rainfall is concentrated in April to September, since the climate is semiarid, with mean annual rainfall of approximately 1,050 mm year [45–47]. The longest and largest tributaries of the Pandeiros River are the Catolé, Suçuarana, Borrachudo and Macaúbas streams.

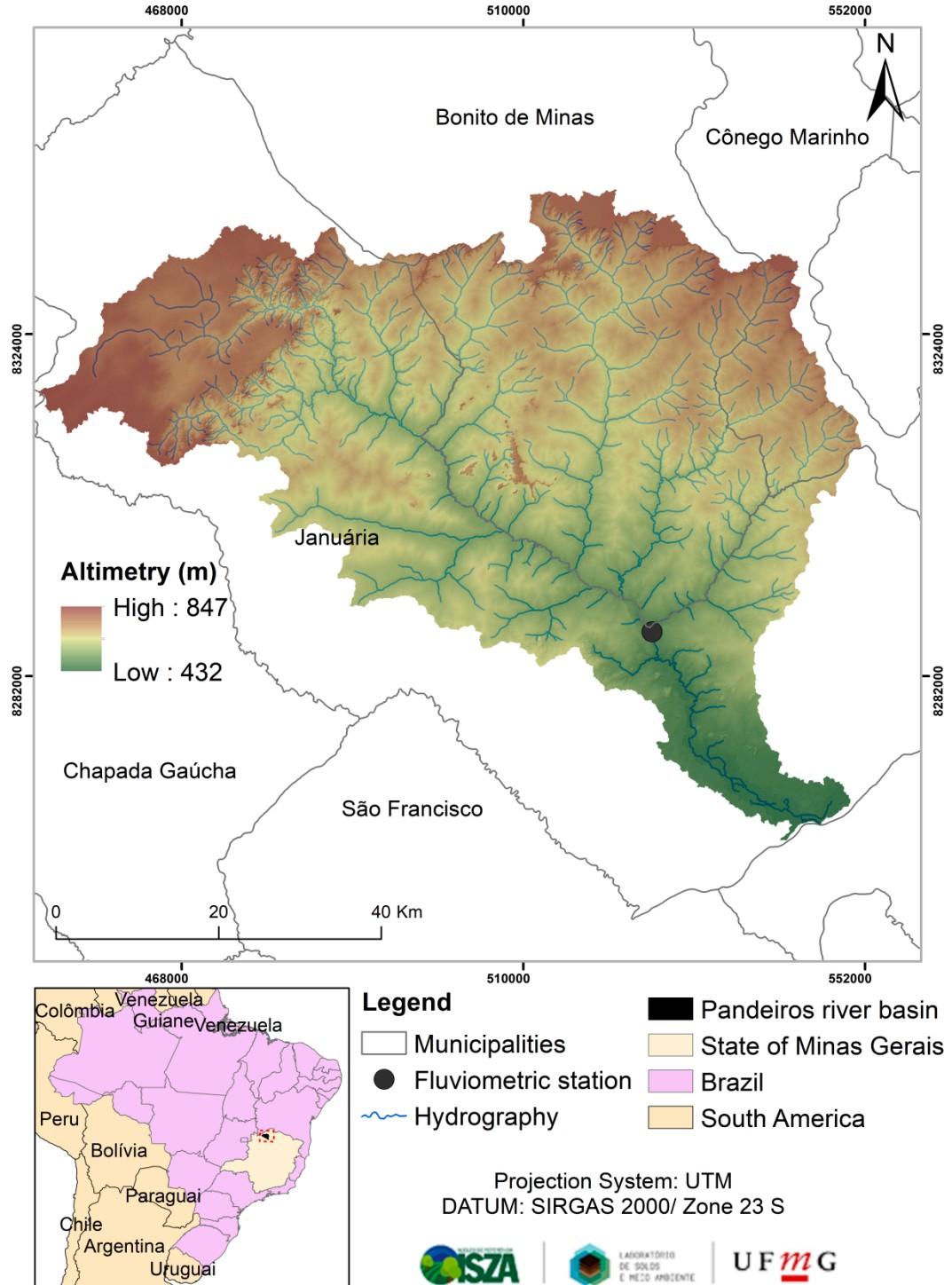

**Figure 1.** Location of the Pandeiros River basin in the state of Minas Gerais, Brazil.

According to the Brazilian Institute for Geography and Statistics (2018), the municipalities comprising the PRB have a total population of 86,311 inhabitants, most of them living in Januária. The largest part of this population live in rural areas, especially in the municipalities of Bonito de Minas and Cônego Marinho (Table 1) [48].

**Table 1.** Area and population living in the municipalities of Pandeiros River basin. Source: Brazilian Institute for Geography and Statistics. The population numbers refer to 2018.

| Municipality | Total Area (ha) | Urban Population (2018) | Rural Population (2018) | Total Population (2018) |
|---|---|---|---|---|
| Januária | 665,700 | 41,322 | 24,141 | 67,628 |
| Bonito de Minas | 390,200 | 2209 | 7464 | 11,088 |
| Cônego Marinho | 164,100 | 1915 | 5186 | 7595 |
| Total | 1,220,000 | 45,446 | 36,791 | 86,311 |

The PRB is in an area with transitional vegetation, presenting phytophysionomies of Cerrado and Caatinga biomes [49]. This ecotone is characterized by swamp regions that contain the springs of the São Francisco River. These springs are responsible for the reproduction of most fishes that live between the Três Marias (MG) and Sobradinho (BA) dams [50].

The relief is predominantly flat. The plain was formed by the filling of the São Francisco Depression with sediments that were sourced from the erosion of rocks from the São Francisco Plateau [51]. This process is also responsible for a small proportion of steep slope areas in the basin, as shown in Figure 2 and quantified in Table 2 [52].

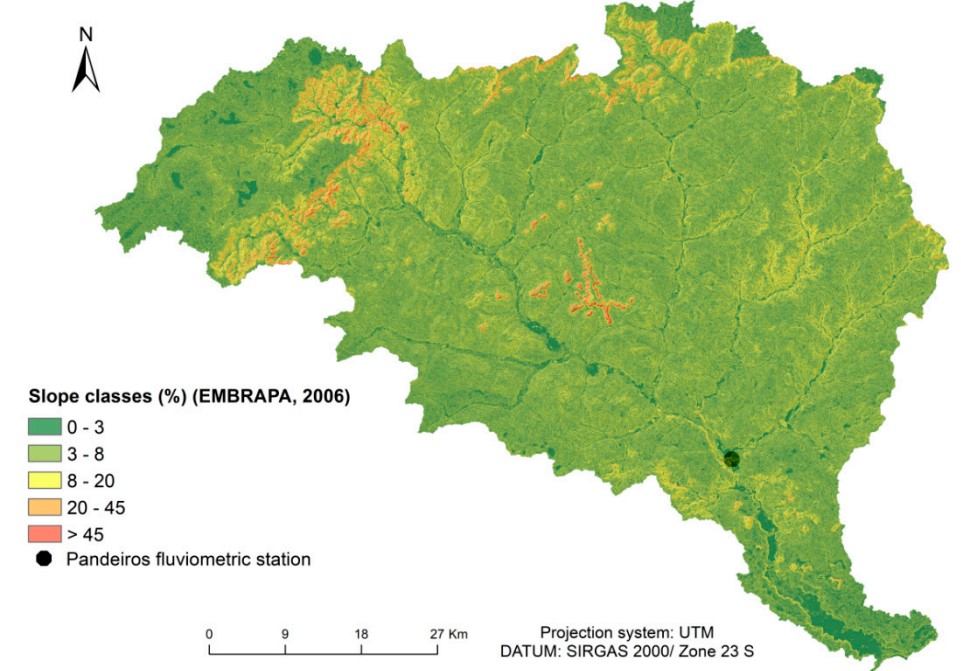

**Figure 2.** Slope map of the Pandeiros River basin.

**Table 2.** Area of slope classes in the Pandeiros River basin.

| Slope (%) | Relief Type | Area (ha) | % |
|---|---|---|---|
| 0 to 3 | Plain | 122.048,53 | 30.83 |
| 3 to 8 | Slightly wavy | 206.106,85 | 52.04 |
| 8 to 20 | Wavy | 59.179,5 | 14.94 |
| 20 to 45 | Strongly wavy | 7.806,56 | 1.97 |
| Above 45 | Mountainous to scarped | 886,85 | 0.22 |
| Total | | 396,028.29 | 100 | 100 |

As regards the local geology, there is the presence of alluvial deposits over the main drains in the area and in wetland regions (Veredas), which are the results of natural and anthropogenic erosion

processes with consequent transport and deposition of sediments. Figure 3 and Table 3 show the spatial distribution and proportion of lithotypes in the studied basin, respectively.

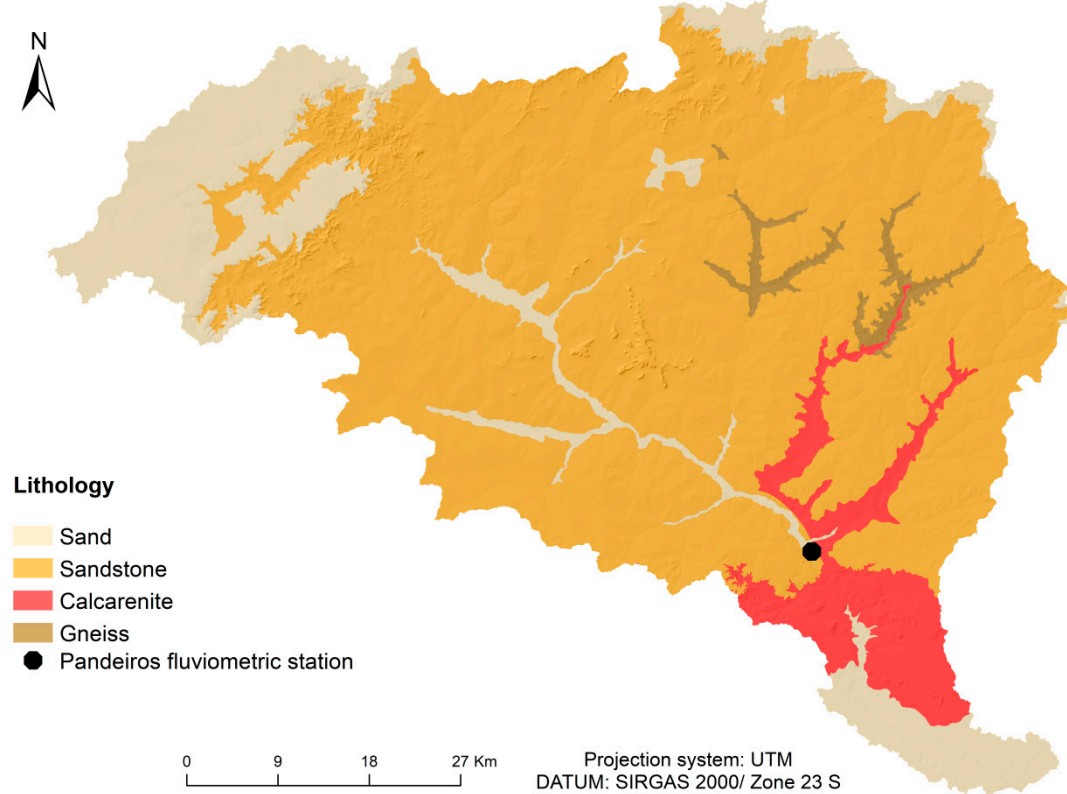

**Figure 3.** Lithological map of the Pandeiros River basin.

**Table 3.** Spatial distribution and proportion of lithotypes in the Pandeiros River basin.

| Lithotype | Area (ha) | % |
|---|---|---|
| Sand | 70,886.74 | 17.90 |
| Sandstone | 290,804.78 | 73.43 |
| Calcarenite | 27,570.98 | 6.96 |
| Gneiss | 6,765.81 | 1.71 |
| Total | 396,028.29 | 100 |

The Soil Map of Minas Gerais [53] presents an area predominantly composed of red–yellow latosols, which cover more than 87% of the hydrographic basin, followed by fluvic neosols (5.48%). The occurrence of quartzarenic neosols, litolic neosols, cambisols and melanic gleysols are related to much smaller areas (Figure 4 and Table 4).

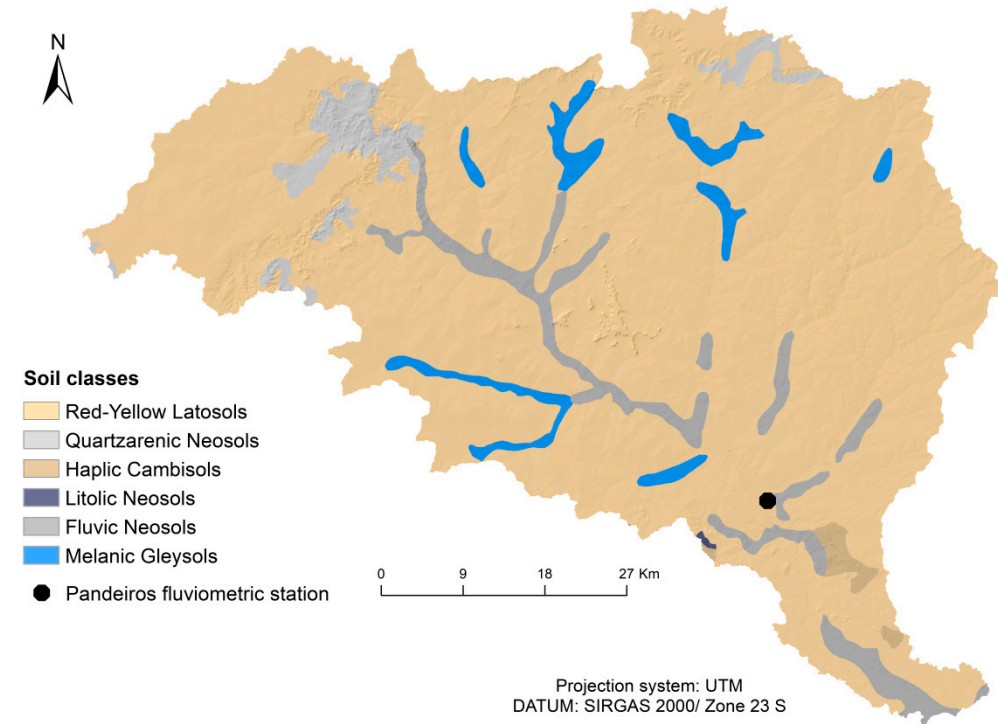

**Figure 4.** Soil map of the Pandeiros River basin.

**Table 4.** Spatial distribution and proportion of soil classes in the Pandeiros River basin.

| Soil Classes | Area (ha) | % |
|---|---|---|
| Haplic cambisols | 2,810.25 | 0.71 |
| Melanic gleysols | 11,856.63 | 2.99 |
| Red–yellow latosols | 346,549.01 | 87.51 |
| Fluvic neosol | 21,706.62 | 5.48 |
| Litolic neosol | 176.09 | 0.04 |
| Quartzarenic neosol | 12,929.70 | 3.26 |
| Total | 396,028.29 | 100 |

The spatial distribution of the land use and cover classes in the basin (Figure 5) shows the predominance of a typical Cerrado vegetation (savanna) of low to medium size [54], which is usually associated with the occurrence of latosols and sandstone regions. These are found mainly in the central part, towards the northern and northeast of the Pandeiros River basin. This phytophysiognomy covers 183,719.88 ha in the basin, representing 46.3% of its area (Table 5). The second most significant land use and cover class in the study area is the dense Cerrado (21.5%), followed by the sparse Cerrado (10.42%).

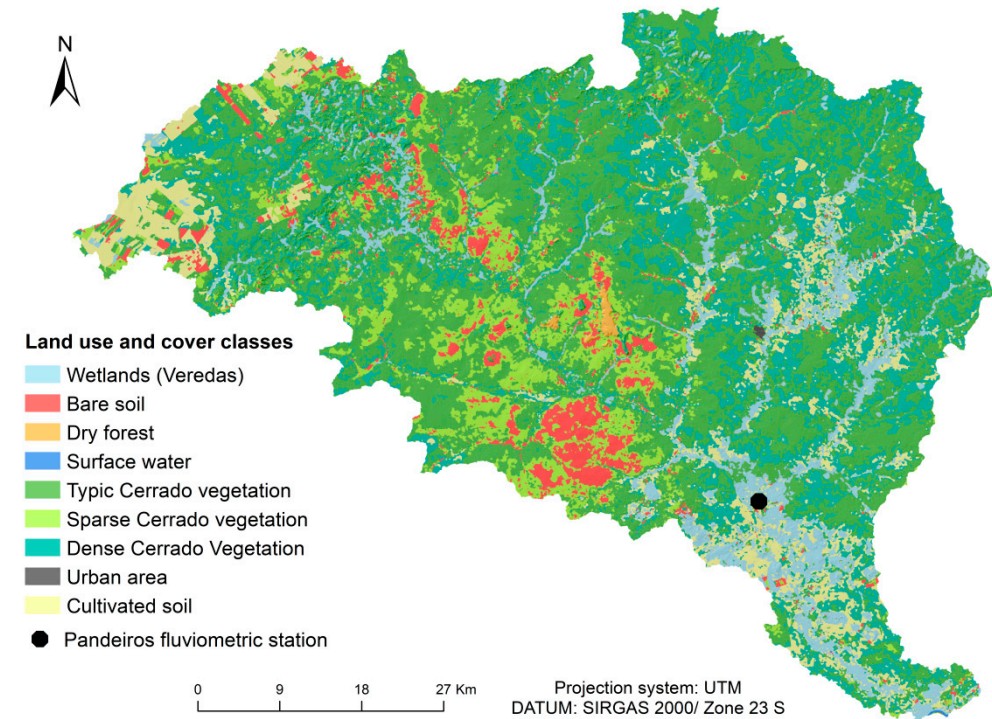

**Figure 5.** Land use and cover map of the study area, Pandeiros River basin.

**Table 5.** Spatial distribution and proportion of the land use and cover classes in the Pandeiros River basin.

| Class | Area | % |
|---|---|---|
| Cultivated soil | 33,138.21 | 8.37 |
| Urban area | 121.19 | 0.03 |
| Dense Cerrado vegetation | 85,203.09 | 21.51 |
| Sparse Cerrado vegetation | 41,246.36 | 10.42 |
| Typical Cerrado vegetation | 183,719.88 | 46.39 |
| Surface water | 70.08 | 0.02 |
| Dry forest | 1,102.50 | 0.28 |
| Bare soil | 18,351.75 | 4.63 |
| Wetland regions (Veredas) | 33,075.23 | 8.35 |
| Total | 396,028.29 | 100 |

*2.2. Material*

The materials used in this study (Table 6) consisted of (i) a digital elevation model (ALOS PALSAR), with a spatial resolution of 12.5 meters; (ii) a land use and cover map (scale of 1: 25.000); (iii) the soil map of Minas Gerais state (scale of 1: 600.000); (iv) values of groundwater recharge calculated by the PUC method and the hydraulic conductivity of each soil class in the basin; (v) rainfall and evapotranspiration data from meteorological stations located in municipalities near the basin; and vi) flow data records from the station code 45,250,000 located near the mouth of the Pandeiros River.

**Table 6.** Material used in the groundwater recharge evaluations.

| Data Type | Use in the Work | Web Site Page |
|---|---|---|
| Digital elevation model | Calculation of slope length and steepness factor | https://www.asf.alaska.edu |
| Land use and cover map | Calculation of RF, the runoff factor | https://www.earthexplorer.usgs.gov/ |
| Soil map | Calculation of PF, the percolation factor | https://www.dps.ufv.br |
| Rainfall and evapotranspiration data | Calculation of recharge potential | https://www.inmet.gov.br |
| Streamflow data | Validation of recharge potential data (analysis of hydrograph recession curve) | https://www.snirh.gov.br/hidroweb |

## 2.3. Methodology

The spatially explicit groundwater recharge potential in the Pandeiros River basin was evaluated using the methodology proposed by Costa et al. (2019) [10], with the adjustments described below. The workflow was divided into 5 main steps: (i) acquisition of a land use and cover map, digital elevation model, soil type map, and climate data (rainfall and evapotranspiration); (ii) calculation of surface runoff using the slope length and steepness factor, and runoff coefficients for the land use and cover types; (iii) calculation of water percolation based on the PUC method [43] (replacing the effective porosity of Costa's approach and representing the proposed methodological improvement) and adapted hydraulic conductivity values with fuzzy logic [55,56]; (iv) calculation of groundwater recharge in different points of the basin and a mean value for the whole basin, using a geographical information system; (v) validation of results by comparing the previously calculated mean groundwater recharge with the value estimated by the hydrograph recession curve analysis (Figure 6).

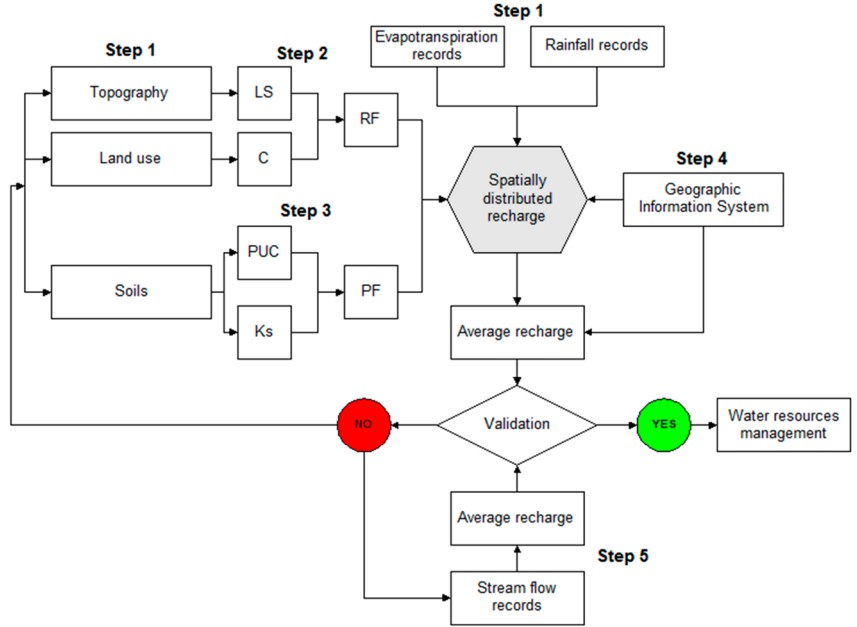

**Figure 6.** Flowchart for the groundwater recharge potential calculation and model validation. Adapted from Costa et al. (2019) [10]. Abbreviations: LS—slope length and steepness factor [57]; C—runoff coefficient (Table 7); RF—runoff factor (Equation (1)); PUC—Conservative Use Potential [43]; Ks—soils' hydraulic conductivity (Table 8); PF—water percolation factor (Equation (2)).

In the first step, the mean annual rainfall and evapotranspiration of municipalities near the study area were estimated using data from the Brazilian Institute for Meteorology (INMET) relative to the 2009–2018 period, which were obtained from meteorological stations in Arinos (MG), Januária (MG), Montes Claros (MG), Salinas (MG), Cariranha (BA), Espinosa (MG), Formoso (MG), Posses (GO), and Brasília (DF). Longer temporal series would be more adequate for estimating groundwater recharge. However, these records were not available. The flaws in the temporal series were resolved by using the regional weighting method, and the information was interpolated through inverse distance weighting (IDW) raised to the power of two [58]. This method was used because it raises the importance of closer stations in the interpolation. The data were spatialized and trimmed up to the study area limits.

In the second step, the land use and cover map and the topographical information from the digital elevation model (slope length and steepness factor—LS factor) were used to calculate the surface runoff factor, based on the method proposed by Böhner and Selige (2002) [59]. The runoff factor was calculated according to Equation (1), reproduced from Costa et al. 2019 [10].

$$RF = 1 - (C + LS_{FUZZY}) \tag{1}$$

where *RF* is the runoff factor (dimensionless); *C* is the runoff coefficient (dimensionless values adopted from Table 7); $LS_{FUZZY}$ is the slope length and steepness factor estimated using the method of Desmet and Govers (1996) [57], but recast to the 0–1 range using a fuzzy logic algorithm (the steeper the slope the closer to 1).

**Table 7.** Runoff coefficients for each land use and cover class in the Pandeiros River basin. Adapted from the ASCE—American Society of Civil Engineers, 1969, and Costa et al., 2019 [10,60].

| Land and Cover Class | Runoff Coefficient |
|---|---|
| Anthropized areas | 0.50 |
| Urban areas | 0.85 |
| Forest formation | 0.10 |
| Exposed soil and ground vegetation | 0.60 |
| Wetland regions ("Veredas") | 0.09 |

In the third step, the water percolation was calculated considering the soil classes in the basin [53]. A systematic literature review was performed to set up hydraulic conductivity values for each soil class, at the second category level of Brazilian soil classification system (SiBCS) [61]. The hydraulic conductivity values were established according to Freire et al. (2003), Costa et al. (2019), Pedron (2011) and Amaral (2017) [10,55,56,62].

**Table 8.** Soil classes and respective $Ks_{Fuzzy}$ values. The scores of groundwater recharge parameter were set up by the PUC method in the Pandeiros River basin [10,55,56,62].

| Soil Class | PUC Score | $Ks_{fuzzy}$ |
|---|---|---|
| Haplic cambisols | 0.4 | 0.3 |
| Melanic gleysols | 0.1 | 0.01 |
| Red–yellow latosols | 0.7 | 1 |
| Fluvic neosols | 0.3 | 0.02 |
| Litolic neosols | 0.3 | 0.06 |
| Quartzarenic neosols | 0.3 | 0.61 |

Different from the method proposed by Costa et al. (2019) [10], the water percolation factor in this model was calculated using the groundwater recharge category of the PUC method, developed by Costa et al. (2017) [43]. The PUC is a method that allows for mapping areas of a basin based on their limitations and potentialities for conservationist land use, through the combined assessment and weighting of several environmental variables (soils, geology, and geomorphology) [43].

The PUC assigns values from 1 to 5 to the different classes of lithology, slope and soil in the watersheds from the state of Minas Gerais, Brazil. The analyses are focused on groundwater recharge, agricultural use potential and soil resistance to erosion in the catchments [43]. For the identification of lithologies, slopes and soil classes existing in Minas Gerais, the available official databases are used.

The attribution of grades for the different types of lithologies took into consideration their potential to provide nutrients (greater weight for rocks with higher absolute content of essential macro elements to plants) and their susceptibility to weathering processes (considering the main mineral constituents and scored according to their resistance to weathering, based on Goldish's stability [63]). Regarding the slope parameter, the same weights were attributed for groundwater recharge potential for agricultural use and resistance to erosion. For groundwater recharge, it was considered that the slope has a direct relationship with the flow velocity and the opportunity time for water infiltration. The higher the slope, the higher the water velocity and the shorter the time for water infiltration. In this context, the mountainous relief received a weight 1 and the flat relief a weight 5.

For the attribution of grades to different soils, the variables texture, drainage, effective depth and fertility were considered. For groundwater recharge, the attribute fertility was disregarded and the classes of soils characterized as favorable to water infiltration and percolation received greater weight.

The recharging potential for each soil type was obtained by the simple average of the values of texture, drainage and effective depth, normalized so that the final scale was in the range of 1 to 5.

In this work, only the soil parameter of groundwater recharge was used, which assigns the basin´s soil classes a score from 1 to 5. This parameter takes into account the effective depth, texture and drainage of each soil class at the first category level of SiBCS [61]. Thus, to the soil that presents considerable effective depth, satisfactory drainage and texture favorable to infiltration, a higher score regarding groundwater recharge is attributed.

However, in order to adapt the PUC method to the recharge model, a rescaling process was implemented, whereby the ratings from 1 to 5 were recast to the range 0 to 1. The values used in the model are represented in Table 8. The percolation factor was evaluated using Equation (2), reproduced from Costa et al. 2019 [10]:

$$PF = PUC \times Ks_{FUZZY} \tag{2}$$

where *PF* is the water percolation factor (dimensionless); $Ks_{FUZZY}$ (dimensionless) is the soil's hydraulic conductivity fitted to the 0 to 1 range by the fuzzy logic algorithm (the higher the soil's hydraulic conductivity in the class the closer to 1); *PUC* are the scores of groundwater recharge set up by the PUC method but fitted to the 0 to 1 range.

In the fourth step, the groundwater recharge potential was calculated for each point in the basin, using Equation (3), according to Costa et al. 2019 [10]:

$$RPot = [(P - ET_R) \times RF \times PF] \times 10 \tag{3}$$

where *RPot* is the groundwater recharge potential (m$^3$ ha$^{-1}$ year$^{-1}$); *P* is the mean annual rainfall depth (mm year$^{-1}$); $ET_R$ is the mean evapotranspiration (mm year$^{-1}$); *RF* is the surface runoff factor; and *PF* is the water percolation factor.

The results were validated in the fifth step through comparison of the calculated mean groundwater recharge with a homologous median value estimated by the hydrograph recession method based on the Maillet equation [64,65]. The hydrograph for the Pandeiros River was drawn from daily average stream flow data measured at the hydrometric station nº 4425000 located in the Pandeiros River mouth (Pandeiros River Dam), compiled from the Hidroweb portal [66]. Streamflow data from 2013 to 2018 were the most recent and more continuous in the historical series, showing no flaws or absent data. Thus, these data were used to calculate the Maillet equation (Equation (4)). To separate and analyze the recession curve and the recession days, the methodology proposed by Barnes (1939), Dewandel et al. (2003) and Kovacs et al. (2005) was used [64,65,67–69]:

$$Q_T = Q_0 \times e^{-\alpha t} \tag{4}$$

where $Q_T$ is the flow at time *t* (m$^3$ s$^{-1}$); $Q_0$ is the flow at the beginning of a recession (m$^3$ s$^{-1}$); $\alpha$ is the coefficient of recession; *t* is the time (days); and *e* is the basis of Neperian logarithm (2.71828).

Thus, the coefficient of recession can be determined numerically, based on the logarithmic form of Equation (4), represented and rearranged in Equation (5):

$$\alpha = \frac{LogQ_0 - LogQ_t}{0.4343t} \tag{5}$$

Subsequently, the groundwater recharge volume was calculated using Equation (6):

$$V = \frac{Q_0 \times t'}{\alpha} \tag{6}$$

where *V* is the recharge volume (m$^3$); $Q_0$ is the flow at the beginning of recession (m$^3$ s$^{-1}$); *t'* is the converter of the *t* unit (days into seconds; 86,400); $\alpha$ is the coefficient of recession (dimensionless).

The constant of recession ($\alpha$; Equation (5)) is dependent on the aquifer characteristics and therefore should not vary significantly from year to year. On a $Q$ versus $t$ plot (hydrograph), where the $Q$ values are represented in logarithmic scale and the values are in linear scale, the baseflow within a hydrologic year (from the recharge period to the end of recession) should define as a straight line, the slope to which is related in terms of $\alpha$. If $t_{cycle}$ is the time of a log cycle for discharge, meaning the time for discharge to change from 1 to 10 m$^3$/s, from 10 to 100 m$^3$/s, and so forth, then $LogQ_0 - LogQ_t = 1$ and $\alpha = 1/(0.4343t_{cycle})$. This simplified representation of Equation (5) is frequently used in the calculation of $\alpha$ and will be adopted in the present study. Conversely, the values of $Q_0$ can vary in response to the annual variations of precipitation. In this case, a value of $Q_0$ should be calculated for each hydrologic year, while mean ± standard deviation values are derived therefrom.

## 3. Results

The interpolation of precipitation data from 2009 to 2018, obtained from climatologic stations close to the Pandeiros River basin, showed mean rainfall depths ranging from 904.7 to 1056.3 mm year$^{-1}$ (Figure 7).

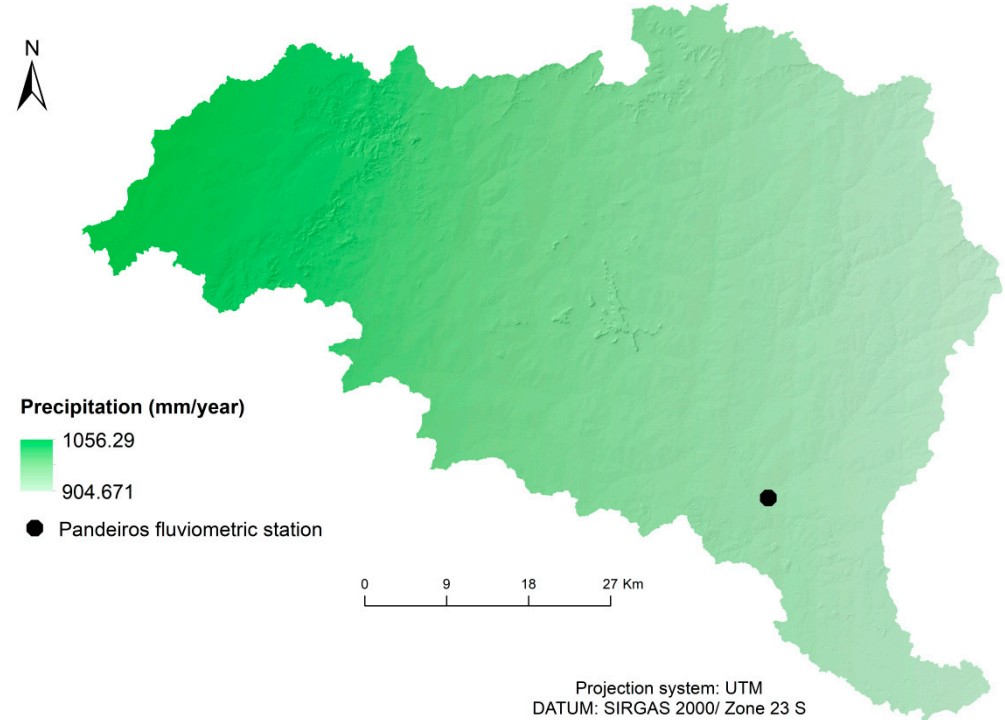

**Figure 7.** Spatial distribution of rainfall in the Pandeiros River basin.

The evapotranspiration ranged from 729.6 to 856.5 mm year$^{-1}$ (Figure 8). The spatial distribution of this variable was similar to rainfall; the lowest values were found in the southeast, east, and northeast regions, whereas the highest values were found in the northwest region of Pandeiros River basins.

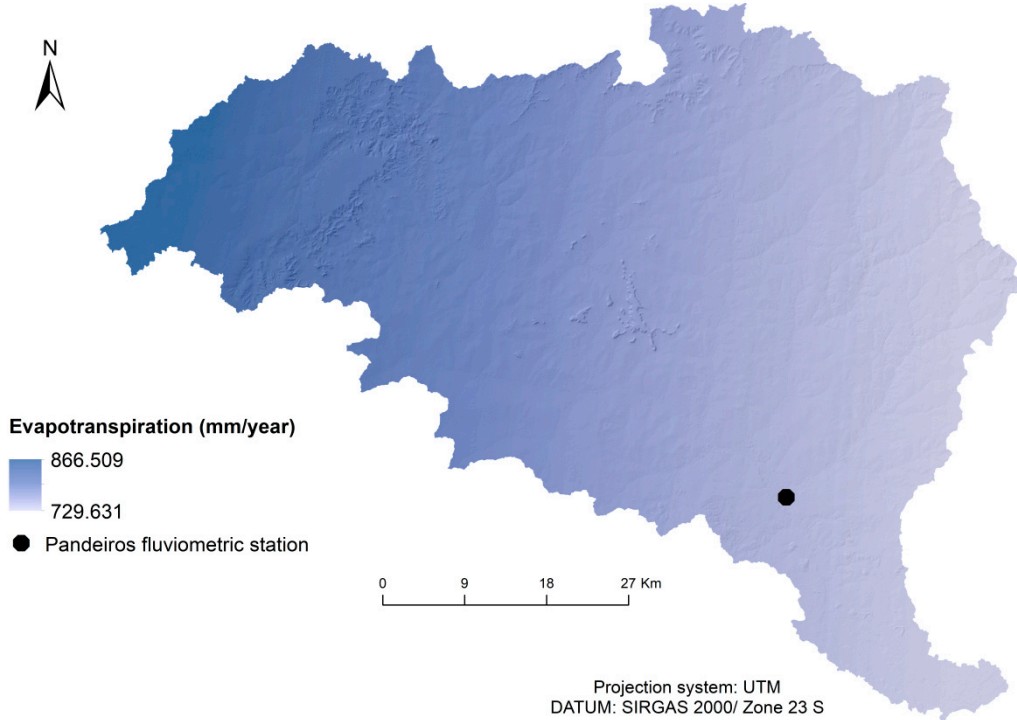

**Figure 8.** Spatial distribution of evapotranspiration in the Pandeiros River basin.

The evaluation of the runoff factor exposes a strong effect of runoff coefficients on the RF, as observed when the land use and cover map (Figure 5) is compared with the runoff map (Figure 9). The map of Figure 9 showed a lower groundwater recharge in urban areas, which have a higher runoff. In contrast, higher potential for groundwater recharge takes place in dense vegetation areas, indicating that the presence of vegetation decreases surface runoff.

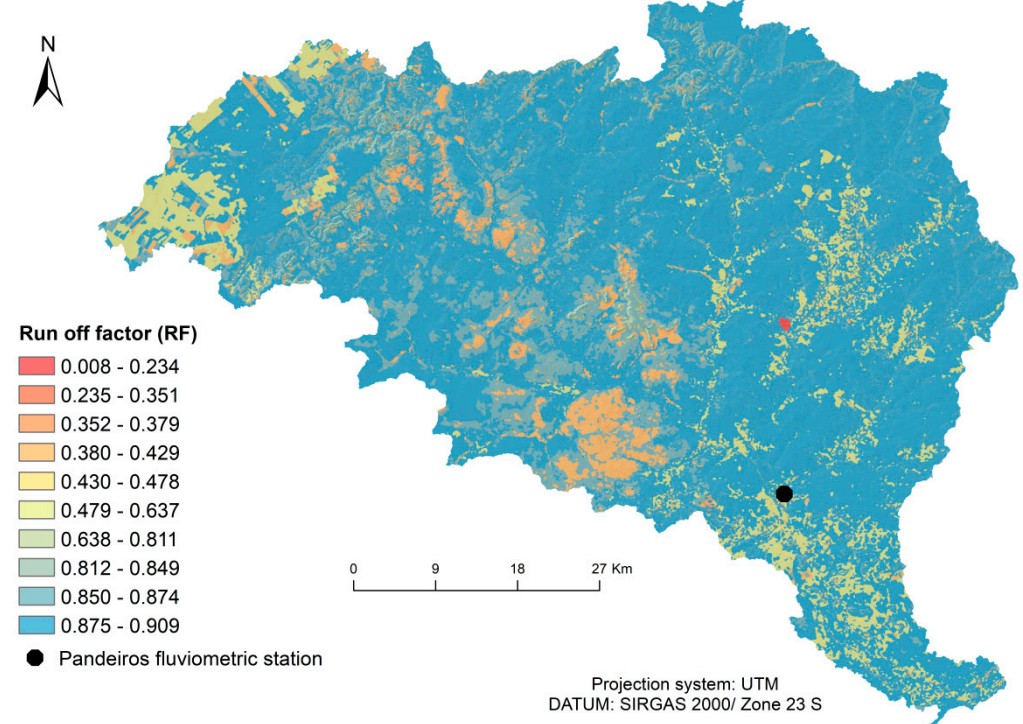

**Figure 9.** Spatial distribution of runoff factor in the Pandeiros River basin.

The water percolation factor ranged from 0.001 to 0.7 (dimensionless), expressing the combined variation of hydraulic conductivity values and PUC scores fitted to the 0–1 range (Figure 10 and Table 8). Areas with melanic gleysols and fluvic neosols presented the lowest water percolation factors, decreasing the groundwater recharge in these areas because of their low hydraulic conductivity.

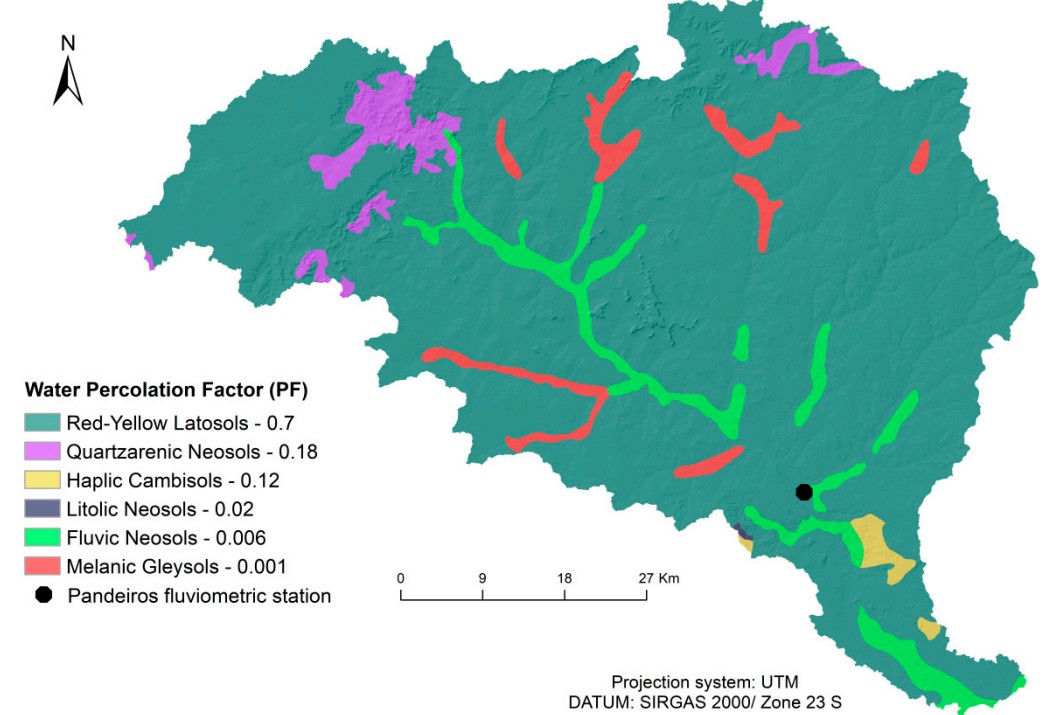

**Figure 10.** Spatial distribution of water percolation in the Pandeiros River basin.

The map of Figure 11 shows the groundwater recharge potential of the Pandeiros River basin ranging from 0 to 122.7 mm year$^{-1}$, with a mean value of 93.99 mm year$^{-1}$. The areas with higher groundwater recharge potential are located in regions with dense tree vegetation cover, areas with flat or slightly wavy relief, and areas with developed and structured soils, where the porosity and hydraulic conductivity allow water percolation to the water table. These areas are distributed throughout the basin and are found in the three municipalities that encompass the basin.

The areas with lowest groundwater recharge potential are represented in Figure 11 with a red color and are located in the urban area and in areas with the presence of melanic gleysols and fluvic neosols. The reduced potential is due to the soil sealing process occurring in urban areas, and to low hydraulic conductivities that decrease water percolation and consequently recharge in the aforementioned soil types.

The validation of PUC-based recharge estimates is depicted in Table 9. The mean groundwater recharge is 93.99 mm year$^{-1}$ which is close to the median value obtained with the hydrograph recession method (87.2 mm year$^{-1}$). The difference between the two values is just 6.6 mm year$^{-1}$ or 7.3%. The hydrograph used to calculate groundwater recharge based on base flows is shown in Figure 12. Hydrographs describe the succession of peaks representing the watershed response to a precipitation event, which are separated by baseflow segments that describe the aquifer response to drainage. Hydrograph recession curves are usually separated in the quick flow stage, which depicts the runoff and water infiltration towards the saturated zone, and the baseflow stage when only the saturated zone discharges [68]. The baseflow discharge is the most representative feature of an aquifer's global response because it is less influenced by the temporal and spatial variations on infiltration [69]. Generally, hydrographs are analyzed together with rainfall data. The peak of a hydrograph rising curve shows the highest values of stream flows, which take place in the months when rainfall values

are the highest. In these periods the superficial runoff also reaches its highest values and it decreases during the recession time, marked by the red segments in Figure 12.

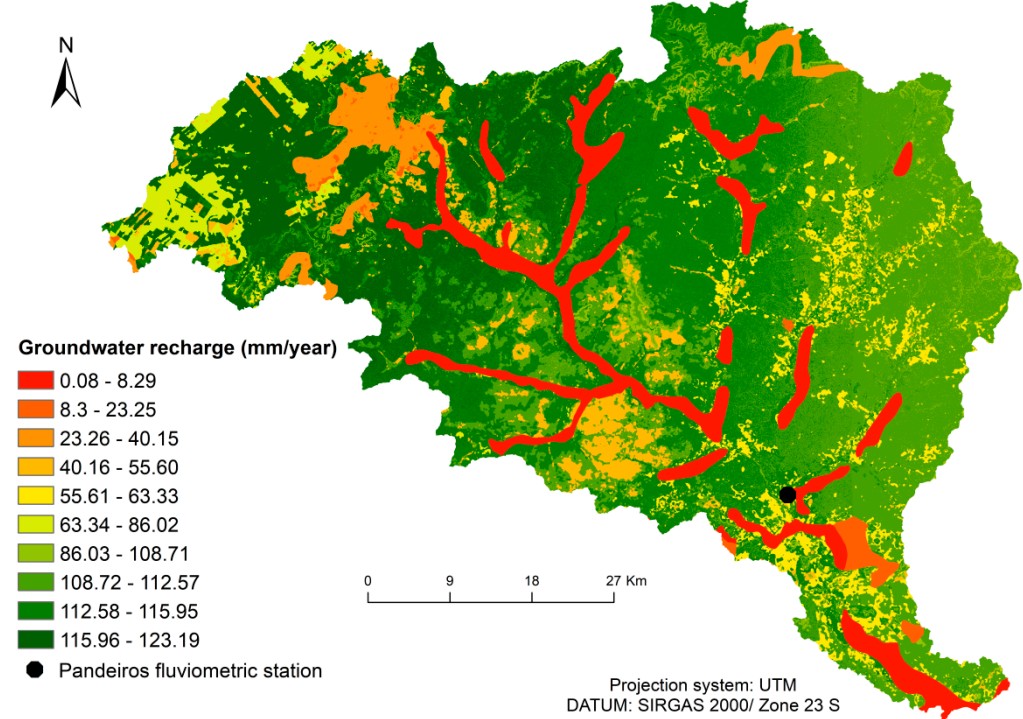

**Figure 11.** Spatial distribution of groundwater recharge potential in the Pandeiros River basin.

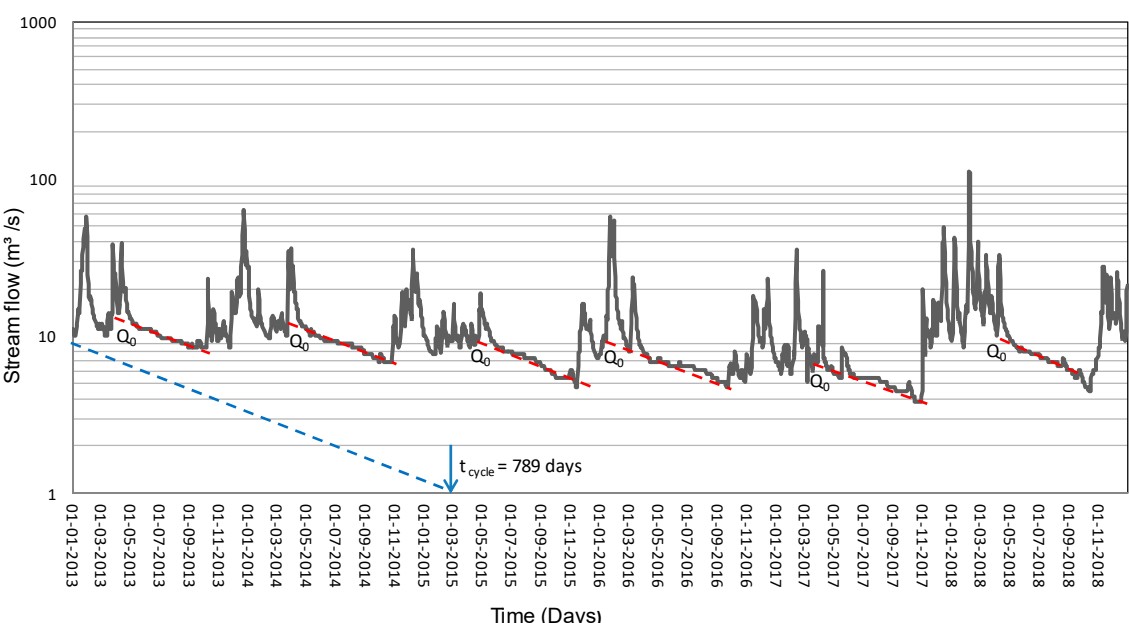

**Figure 12.** Hydrograph of Pandeiros River basin, state of Minas Gerais, Brazil, used to calculate the recession constant value.

**Table 9.** Groundwater recharge in the Pandeiros River basin, estimated by the spatially distributed PUC-based and hydrograph recession analysis methods.

| Method | Mean Groundwater Recharge (mm year$^{-1}$) | Average Difference (mm year$^{-1}$) | Average Difference (%) |
|---|---|---|---|
| Spatialization | 93.99 ± 36.92 | 4.8 | 5.2% |
| Hydrograph recession curve analysis | 98.77 ± 20.32 | | |

In the assessment of recharge using the method of hydrograph recession analysis, we looked for straight line segments corresponding to base lows. These segments allowed us to draw the corresponding fitting lines (dashed red lines). As would be expected, the lines are all virtually parallel because the line slope is solely dependent on the aquifer characteristics and dimension, which are invariant at the timescale of a few years (present case). For all years, the point in the graph where the fitting line intercepted the hydrograph at the upper flows (left edge point) was defined as the $Q_0, t_0$ point, i.e., the point where the recession period began. On average $Q_0 = 10.8 \pm 1.8$ m$^3$ s$^{-1}$. The coefficient of recession was estimated by the simplified version of Equation (5), $\alpha = 1/(0.4343 t_{cycle})$. In the Pandeiros River basin, the estimated $t_{cycle}$ is 789 days, and therefore $\alpha = 0{,}002918$. Using $Q_0$ and $\alpha$ in Equation (6) results in the recharge value of $V = 98.8 \pm 20.3$ mm year$^{-1}$.

Figure 13 shows the spatialization of groundwater recharge within the municipalities that constitute the studied basin. Although the municipality of Januária encompasses a large area of the Pandeiros River basin, approximately 212,956.8 ha, this municipality presents a higher proportion of melanic gleysols, fluvic neosols, and litolic neosols, which are soils with the lowest water percolation factor (PF) assessed by the model. Moreover, it presents a considerable proportion of exposed soil, sparse Cerrado vegetation, and cultivated soils in the northwest region of the basin, which increases the runoff potential. For these reasons, this municipality presented a mean annual groundwater recharge of 90.30 mm year$^{-1}$.

The municipality of Cônego Marinho presented the highest mean annual groundwater recharge (105 mm year$^{-1}$). This municipality occupies a smaller area of the basin (approximately 26,083.86 ha) than the other municipalities, but it is in a region where soil and topography favor groundwater recharge.

Bonito de Minas presented mean annual groundwater recharge of 97.18 mm year$^{-1}$. The area of the basin within this municipality is approximately 156,987.95 ha. Despite the considerable presence of fluvic neosols and melanic gleysols in this region of the basin, the proportion is lower than that of Januária. The urban area of Bonito de Minas within the basin is small, representing 0.03% of the total area of the basin, which does not significantly affect the mean annual groundwater recharge.

The land use and cover map (Figure 5) and the groundwater recharge map (Figure 11) showed that the areas with forest cover presented the best recharge values. This type of land use and cover combined with flat or slightly wavy relief and areas overlaid with red–yellow latosols proved preferable for groundwater recharge.

Contrastingly, regions presenting such land use and cover as urban areas, exposed soil, ground vegetation, poorly structured soils and low hydraulic conductivity (fluvic neosols, litolic neosols, haplic cambisols, melanic gleysols), combined with steeper areas, presented lower recharges.

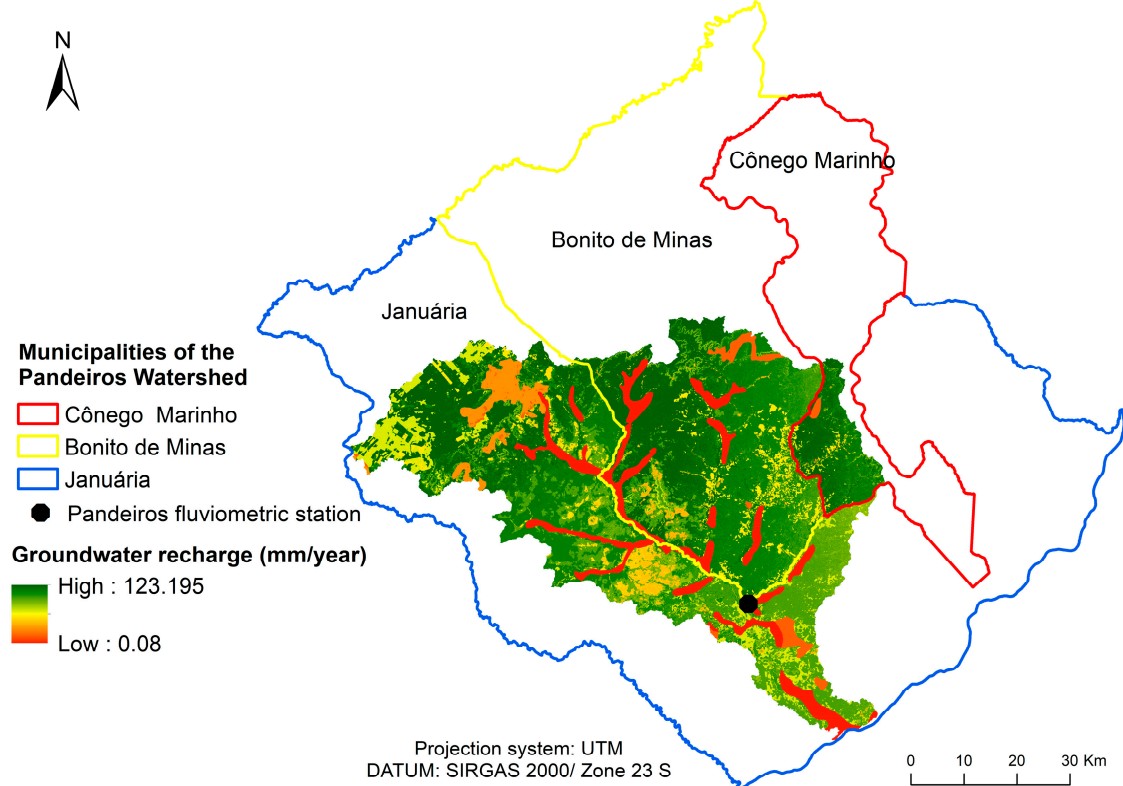

**Figure 13.** Spatial distribution of groundwater recharge potential in the municipalities of Pandeiros River basin.

## 4. Discussion

The rainfall depths (Figure 7) are consistent with Neves (2011), who evaluated a historical series (1931–1990) in the same region and reported a mean rainfall depth of 1050 mm year$^{-1}$ [45]. However, the Brazilian Institute for Geography and Statistics (2014) considers values below 800 mm year$^{-1}$ to define the semiarid region of Brazil [70], denoting that the rainfall depths of the study area were relatively high for a semiarid region.

Yang et al. (2016) showed that the rainfall depth can be a determinant for evapotranspiration and temperature values [71]. Therefore, practically all water that leaves the system (output) in regions with dry climate and low rainfall, such as the Brazilian semiarid regions, is due to evapotranspiration [72]. According to Loos, Gayler, and Priesack (2007), evapotranspiration is one of the most critical variables, with a high impact on water loss in similar regions [73].

The spatial distribution of runoff factor showed that cover plants are essential to maintaining the water cycle and to protect the soil against the impacts of raindrops. Moreover, the presence of vegetation increases soil porosity and permeability by the action of roots, thereby reducing runoff, and keeping soil moisture in the vicinity of organic colloids [74].

Soils under forest are characterized by expressive plant residue layers (litter fall) and by an A horizon rich in organic matter, which enables a higher soil aggregation, preserving its porosity [75]. Soils under forest usually present significant porosity, mainly macropores due to dead roots and animal holes, which are important to facilitate water infiltration and recharge. Therefore, water infiltration capacity is usually more expressive in areas with forest vegetation [75,76] than in pastureland or cropland, as found in the present work, resulting in a lower surface runoff.

Costa et al. (2019) [10] used a groundwater recharge model and reported a higher surface runoff in urban areas than in vegetated areas. Urban areas have drainage systems, street pavements, and infrastructures that hinder infiltration of rainwater into the soil, decreasing the recharge potential [10].

Comparing the first model employed by Costa et al. (2019) with the present application, it is interesting to see how including the groundwater recharge parameter from the PUC method improved the spatialization of groundwater recharge, mainly because the first model [10] used solely total porosity values for each soil class in the Jequitiba River basin [10]. This parameter is not the best to assess drainage and water percolation, because according to Silva et al. (2013), the micro and macro porosity better reflect the movement of water through the soil profile [77]. Moreover, the PUC method uses a management approach that was not taken into consideration in the PF in the first model. Another important difference from the first model to the present one is the separation until the second category level of the neosols class, which has shown more reliable spatialization and values for the different category levels of this soil class. The differences between the physical characteristics (structure, texture and effective depth) of fluvic neosols, litolic neosols and quartzarenic neosols) make it questionable to give the same values of hydraulic conductivity and porosity to this soil class.

Regarding the validation stage of the work, the higher the slope of the recession curve, the faster the depletion of the water table reserves and the higher the demand of this system to regulate the surface system; whereas the lower the slope, the lower the time of depletion of water resources. The surface and groundwater form a system with mutual contribution, thus, any change in one will affect the other in the short or long term.

The comparison of the methods (spatialization; recession curve analysis) showed a small difference (5.2%) because of their particularities, since each method has inherent and inevitably uncertainty levels [78]. The spatialization method probably has a higher uncertainty in groundwater recharge values because the number of parameters involved with the calculations is much larger [10]. However, the recession curve analysis provides only an average groundwater recharge estimate (mean) for the entire basin, whereas the spatialization method provides one estimate for each point in the basin.

The small difference between the results obtained with the spatially distributed PUC-based and hydrograph recession methods attributes to geology a limited role in the estimation of recharge in the Pandeiros River basin, and that this process is predominantly controlled by the infiltration capacity and profile depth of the soil. The effect of vadose zone thickness on the recharge was recently reported in China [79]. Information on the characteristics of a river basin and its potentialities and limitations are essential to an adequate management of water resources [10,80,81].

The potentialities found for the municipality of Cônego Marinho include the prevalence of red–yellow latosols, which are soils that favor groundwater recharge because of their structural characteristics, such as the occurrence of macropores that increase hydraulic conductivity [43]. Moreover, the region has few cultivated areas and has a predominance of typical Cerrado vegetation, which also favor groundwater recharge, according to the model. Thus, the mean annual groundwater recharge was higher in this municipality than in the other two in the basin.

The considerable overlay of fluvic neosols and melanic gleysols in Bonito de Minas is compensated for by a high proportion of typical Cerrado and dense Cerrado vegetation and by the presence of few cultivated areas, which decreased the runoff potential: Consequently, the mean annual water recharge in this municipality was higher than that in Januária. The presence of native vegetation favors the groundwater recharge, because the forest reduces the surface runoff, favors water percolation, and maintains the soil physical and mechanical stability, assisting in the storage of water and the supply of groundwater [10]. Several studies evaluated the effect of land use and cover and the benefits of forested areas to groundwater recharge [42,82–88].

Considering these results, the method used in the present work provides aid in the better management of river basins by considering their potentialities and limitations, not generalizing a mean value for the whole area evaluated. Therefore, preferred areas can be identified for recharge, thus directing public policies and conservationist actions for each area, according to their needs.

## 5. Conclusions

The groundwater recharge potential was higher in areas covered with forests, located in plains or slightly wavy relief areas, or overlaid with red–yellow latosol. These are the areas to protect in the watershed management plan if recharge is to be favored or restored. The soil classes and their structural attributes, as well as the land use and cover types were considered as key factors for groundwater recharge. The results showed that areas with higher groundwater recharge potential were concentrated in the municipality of Cônego Marinho, followed by Bonito de Minas, and Januária. Areas with a presence of melanic gleysols and fluvic neosols presented the worst responses in the model.

As made evident from the results, this study should be used as a tool for the management of water resources in the Pandeiros River basin, because the preferred recharge areas could be successfully identified. It is urgent therefore that public policies and conservationist actions are enforced in these areas to improve natural groundwater recharge, and hence increase the accessible water volume to the local population. The adjustment of irrigation methods, adoption of soil preservation practices to improve water infiltration, seasonal storage of surface water in areas of low recharge potential and the preservation of forest vegetation, are examples of feasible actions. Moreover, this work provided subsidies for further studies that seek methods for the spatialization of groundwater recharge potential in river basins with a key role assigned to management practices.

**Author Contributions:** Conceptualization, M.A.T., M.S.d.M. A.M.d.C., P.K.M., J.H.M.V.; methodology, M.A.T., M.S.d.M., A.M.d.C. and J.H.M.V.; validation, M.A.T and F.A.L.P.; resources, A.M.d.C.; writing—original draft preparation, M.A.T. and M.S.d.M.; writing—review and editing, M.A.T., M.S.d.M., A.M.d.C., P.K.M., F.A.L.P and L.F.S.F.; supervision, A.M.d.C.; funding acquisition, A.M.d.C. All authors have read and agreed to the published version of the manuscript.

**Funding:** This study was funded by the research project "Sustentabilidade da Bacia do Rio Pandeiros" (Sustainability of Pandeiros River Basin), sponsored by grant APQ-03773-14 from FAPEMIG – Fundação de Amparo à Pesquisa do Estado de Minas Gerais that included a research scholarship for the author Maíse Soares de Moura. The manuscript translation from Portuguese to English was financed by the CAPES – Coordenação de Aperfeiçoamento de Pessoal de Nível Superior. For the author integrated in the CITAB research centre, the research was financed by National Funds of FCT–Portuguese Foundation for Science and Technology, under the project UIDB/04033/2020. For the author integrated in the CQVR, the research was supported by National Funds of FCT–Portuguese Foundation for Science and Technology, under the projects UIDB/00616/2020 and UIDP/00616/2020.

**Acknowledgments:** This study was conducted within the work plan of research group "GEISS – Grupo de Estudos Integrado em Solos e Sustentabilidade", in its research line on "Gestão Sustentável de Recursos Hídricos e Segurança Hídrica". We wish to thank the support of the team of Soil and Environment Laboratory of the Federal University of Minas Gerais for the help in the analyses made in the study.

**Conflicts of Interest:** The authors declare no conflict of interest. The funders had no role in the design of the study; in the collection, analyses, or interpretation of data; in the writing of the manuscript or in the decision to publish the results.

## Nomenclature

The list of mathematical symbols and their measurement units as used in the present article are listed below in alphabetical order:

| | |
|---|---|
| $A$ | recession coefficient; |
| $C$ | runoff coefficient (dimensionless); |
| $E$ | basis of Neperian logarithm (2.71828) |
| $ET_R$ | mean evapotranspiration (mm year$^{-1}$); |
| $Ks_{FUZZY}$ (dimensionless) | soil hydraulic conductivity fitted to the 0 to 1 range by the fuzzy logic algorithm; |
| $LS_{FUZZY}$ | slope length and steepness factor; |
| $P$ | mean annual rainfall depth (mm year$^{-1}$); |
| $PF$ | water percolation factor (dimensionless); |
| $PUC$ | Conservative Use Potential; |
| $Q_T$ | stream flow discharge at time $t$ (m$^3$ s$^{-1}$); |
| $Q_0$ | stream flow discharge at the beginning of a recession (m$^3$ s$^{-1}$); |
| $RF$ | runoff factor (dimensionless); |
| $RPot$ | groundwater recharge potential (m$^3$ ha$^{-1}$ year$^{-1}$); |
| $T$ | time (day); |
| $t'$ | converter of $t$ measurement unit (days into seconds; 86,400) |

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
