# Peer review of "An Improved Model for the Evaluation of Groundwater Recharge Based on the Concept of Conservative Use Potential: A Study in the River Pandeiros Watershed, Minas Gerais, Brazil"

_water, doi:10.3390/w12041001_

Round 1

Reviewer 1 Report

This study evaluates the groundwater recharge potential of the River Pandeiros, Minas Gerais, Brazil. The authors justify the importance to map groundwater recharge potential areas at a catchment scale. The proposed method applied the water recharge category of the Conservative Use Potential (PUC) method and the recharge calculations by means of a water balance by Costa et al. (2017) and Costa et al., (2019), respectively.

In general, the study is well designed and explained, and shows the necessity of the study for the local application. I suggest that the paper be accepted with some minor revisions as follows:

- Study objectives: References are missing to justify the approach (line 44-45, line 68-69)

- Introduction: The statement in line 57-59 needs a better reference.

- Materials and Methods:

  • Please add the location of the hydrological station 45250000, which was used for a validation of recharge potential, on any of Figures 1-5.
  • Figure 6: Abbreviation labels need foot-note explanation, since explanatory texts came afterward. One arrow from the variable ‘Evapotranspiration records’ lacks corresponding variable.

- Results:

  • Table 7: Please confirm the information is regarding soil or land in the ‘soil use and cover class’. Same for the text in line 288, which states ‘soil use and cover map (Figure 5)’.
  • Line 349: Wrong citation to Figure: ‘The land use and cover map (Figure 6)’

- Discussions:

  • Line 387: Please check the reported data is 16% or 5.18% as in Table 8 (line 320).

Author Response

Reviewer #1

 This study evaluates the groundwater recharge potential of the River Pandeiros, Minas Gerais, Brazil. The authors justify the importance to map groundwater recharge potential areas at a catchment scale. The proposed method applied the water recharge category of the Conservative Use Potential (PUC) method and the recharge calculations by means of a water balance by Costa et al. (2017) and Costa et al., (2019), respectively.

 In general, the study is well designed and explained, and shows the necessity of the study for the local application. I suggest that the paper be accepted with some minor revisions as follows:

 Many thanks for the general positive appreciation on our manuscript. We will do our utmost to improve it in keeping with your comments and suggestions. All the changes to the manuscript are highlighted in yellow in the revised manuscript. Below are the answers to your comments.

- Study objectives: References are missing to justify the approach (line 44-45, line 68-69)

In lines 44-45 we added 5 more references included in an additional sentence, as follows:

“It has been estimated that 1/3 of all countries will have to adapt their productive processes up to 2025 because of the lack of water, and more than 3 billion people might be living in regions under chronic draughts or water stress [2–6]. Owing to its great importance in society, economy and the environment, it is necessary to properly manage water resources. The increasingly degradation of surface water resources not only in Brazil but all over the world [7], makes the sustainable management of groundwater become an even more essential activity. In this context, adequate distribution of groundwater requires accurate comprehension and quantification of the input into groundwater systems. This quantification is known as groundwater recharge estimation. In the light of this, studies and maps of areas related to the provision of water resources are needed, with the objective of indicating priority areas for conservation or restoration and direct public policies to protect this natural resource. Studies in Brazil are concentrated mostly on river basin level, which can be defined as the land drained by a river and its tributaries that are formed in the higher regions, draining streams/rivers or infiltrating water into the soil originating springs [8].”

In lines 68-69 we added four references to the mentioned sentence, as follows:

    “ Despite the importance of sustainable management of aquifer recharge zones [7,21–23], in the state of Minas Gerais, Brazil, this topic has not been properly studied.”

- Introduction: The statement in line 57-59 needs a better reference.

We added two more better references to the statement, as follows:

“Changes in the hydrological dynamics are mainly worrying in aquifer recharge zones, which are regions that enable water infiltration and percolation towards an aquifer system, which is defined as geological system capable of storing and distributing a significant amount of water [13–15].”

- Materials and Methods:

  • Please add the location of the hydrological station 45250000, which was used for a validation of recharge potential, on any of Figures 1-5.

The hydrologic station’s location  was added to Figures 1 to 5 as requested.

  • Figure 6: Abbreviation labels need foot-note explanation, since explanatory texts came afterward. One arrow from the variable ‘Evapotranspiration records’ lacks corresponding variable.

Figure 6 was corrected and the abbreviations were explained in the figure caption.

- Results:

  • Table 7: Please confirm the information is regarding soil or land in the ‘soil use and cover class’. Same for the text in line 288, which states ‘soil use and cover map (Figure 5)’.

The information refers to land. Information in Table 7 and in line 288 from the original manuscript was updated in that regard.

  • Line 349: Wrong citation to Figure: ‘The land use and cover map (Figure 6)’

The reference to the land use and cover map was corrected (Figure 6) was replaced by (Figure 5).

- Discussions:

  • Line 387: Please check the reported data is 16% or 5.18% as in Table 8 (line 320).

The difference is 16%. Data in Table 8 was mistaken and were corrected in the revised version.

Reviewer 2 Report

Thank you for the opportunity to review the manuscript entitled "An improved model for the evaluation of groundwater recharge based on the concept of Conservative Use Potential - A study in the hydrographic basin of River Pandeiros, Minas Gerias, Brazil." This manuscript is focused on modifications to an existing model (PUC) to improve understanding of groundwater recharge by modifying the soil parameters to include land management of soils as a more applicable factor in modeling groundwater infiltration. Conceptually, this paper is very interesting and may provide some novel insights into the relationship between land use and aquifer recharge, however it requires significant revision to reach that.

Throughout the manuscript

  • There are significant language and style edits that need to occur to ensure the reader fully understands the importance of this work.
  • All figures need larger text, Figure 1 needs a reference map to show readers where Minas Gerias is relative to the rest of Brazil and the surrounding countries.

Introduction

  • There are a lot of models that exist to examine groundwater recharge in various environments. I would love to see a clearer discussion of this literature before focusing on the PUC model.

Methods/Results

  • I would like to see a full description of the original PUC model included in the methods (as well as in the results so that the reader can fully understand the impact of the modified model relative to the original).
  • The authors discuss a number of calculations that were made but do not include the equations used. This may potentially be addressed by addressing the above comment as well, I am unclear what calculations are part of the PUC model and what are addressing the modifications to the model.
  • I think the authors need to review the application of Malliet's equation. As presented in the results, it appears that they have applied it incorrectly. Figure 12 in particular suggests that the authors are not following the established methodology (they need to identify straight line segments of the recession curve and only apply Malliet to that segments). A number of papers can give the authors a better idea of how to apply these methods. A few examples:
    • Dewandel, B, Lachassagne, P, Bakalowicz, M, Weng, P, Al-Malki, A. 2003. Evaluation of aquifer thickness by analyzing recession hydrographs. Application to the Oman ophiolite Hard-rock aquifer. Journal of Hydrology, 274: 248-269.
    • Kovacs, A, Perrochet, P, Kiraly, L, Jeannin, P. 2005. A Quantitative Method for Characterization of Karst Aquifers Based on Spring Hydrograph Analysis. J. of Hydrology, 303: 152-164.

Discussion/Conclusion

  • English language and style editing may help this section considerably.
  • I would love to see the discussion focus on the comparison to the original PUC model as well as to other models that exist.

Author Response

Reviewer #2

Thank you for the opportunity to review the manuscript entitled "An improved model for the evaluation of groundwater recharge based on the concept of Conservative Use Potential - A study in the hydrographic basin of River Pandeiros, Minas Gerias, Brazil." This manuscript is focused on modifications to an existing model (PUC) to improve understanding of groundwater recharge by modifying the soil parameters to include land management of soils as a more applicable factor in modeling groundwater infiltration. Conceptually, this paper is very interesting and may provide some novel insights into the relationship between land use and aquifer recharge, however it requires significant revision to reach that.

 Many thanks for the general positive appreciation on our manuscript. We will do our utmost to improve it in keeping with your comments and suggestions. All the changes to the manuscript are highlighted in yellow in the revised manuscript. Below are the answers to your comments.

Throughout the manuscript

  • There are significant language and style edits that need to occur to ensure the reader fully understands the importance of this work.

The English was revised throughout the manuscript.

  • All figures need larger text, Figure 1 needs a reference map to show readers where Minas Gerias is relative to the rest of Brazil and the surrounding countries.

All figures were adjusted for font size. Figure 1 was further adjusted to provided the readers the location of Minas Gerais and identification of Brazil’s surrounding countries.

Introduction

  • There are a lot of models that exist to examine groundwater recharge in various environments. I would love to see a clearer discussion of this literature before focusing on the PUC model.

Thanks for this pertinent comment. We followed the suggestion and added the following paragraph to the revised version of the manuscript

“Numerous groundwater recharge estimation methods exist. However, the scientific literature indicates that these methods are associated with a certain degree of uncertainty [25]. These limitations occur mainly because of the lack of available hydrological and hydrogeological data and because of spatial and temporal variation in recharge values [26]. This difficulty is significant for semi-arid areas [27], causing recharge estimations in these areas to be even more challenging to calculate. There are direct and indirect methods to evaluate the groundwater recharge potential. The direct methods include geological and geophysical explorations, gravimetrical and magnetic models, and perforation tests [10]. The indirect methods include hydrological and hydrogeological models [28,29], using geographical information systems (GIS) combined with field works [30,31]. Other studies have employed different methods to estimate groundwater recharge including tracer methods, water table fluctuation methods, lysimeter methods and simple water balance techniques. Some of these studies have used numerical groundwater models or dynamically link it to hydrological models to estimate variations under different climate and land cover conditions [32–36]. For example, Döll (2008) modeled global groundwater recharge using the WaterGAP Global Hydrological Model (WGHM), which has failed to reliably estimate recharge in semi-arid regions [37]. In that study, the influence of vegetation was not taken into account, even though many studies have showed the importance of this variable to estimate the groundwater recharge [32,38–42]. Moreover, Chowdhury et al (2010) delineated water recharge zones in West Medinipur district, India, using an GIS approach mixed with remote sensing and multi-criteria decision making techniques [22]. The input variables considered in that study were geomorphology, geology, drainage density, slope and aquifer transmissivity. Notwithstanding, it is valuable to add that to determine which method is the best for calculating the groundwater recharge is necessary to realize a broad survey on the literature about this issue. The choice of a method should consider the precision level needed, the project execution viability, and available financial resources.”

Methods/Results

  • I would like to see a full description of the original PUC model included in the methods (as well as in the results so that the reader can fully understand the impact of the modified model relative to the original).

Many thanks for this very pertinent suggestion. Please see the paragraph below where more detailed description of the PUC model is provided. This paragraph was added to the revised version of the manuscript.

“The PUC assigns values from 1 to 5 for the different classes of lithology. slope and soil untill the first categorical level of the Brazilian Soil Classification System [59] in the watersheds of the state of Minas Gerais, Brazil. The analyses are focused on water recharge, agricultural use potential and soil resistance to erosion in the watershed [43]. For the identification of the lithologies, slope and soil classes existing in Minas Gerais, the available official databases were used.

The attribution of grades for the different types of lithologies took into consideration their potential to provide nutrients (greater weight for rocks with higher absolute content of essential macroelements to plants) and their susceptibility to weathering processes (considering the main mineral constituents and scored according to their resistance to weathering, based on Goldish's stability [61]). Regarding the slope parameter, the same weights were attributed for water recharge, potential for agricultural use and resistance to erosion. For water recharge, it was considered that the slope has a direct relationship with the flow velocity and the opportunity time for water infiltration. The higher the slope, the higher the water velocity and the shorter the time of water infiltration, thus, the mountainous relief received weight 1 and the flat relief, weight 5.

For the attribution of grades of different soils for each of the three analyses, the variables: "texture", "drainage", "effective depth" and "fertility" were considered. For water recharge, the attribute "fertility" was disregarded and the classes of soils characterized as favorable to water infiltration and percolation received greater weight.  The recharging potential for each soil was obtained by the simple average of the values of texture, drainage and effective depth, normalized so that the final scale was in the range of 1 to 5.”

  • The authors discuss a number of calculations that were made but do not include the equations used. This may potentially be addressed by addressing the above comment as well, I am unclear what calculations are part of the PUC model and what are addressing the modifications to the model.

  • I think the authors need to review the application of Malliet's equation. As presented in the results, it appears that they have applied it incorrectly. Figure 12 in particular suggests that the authors are not following the established methodology (they need to identify straight line segments of the recession curve and only apply Malliet to that segments). A number of papers can give the authors a better idea of how to apply these methods. A few examples:

Dewandel, B, Lachassagne, P, Bakalowicz, M, Weng, P, Al-Malki, A. 2003. Evaluation of aquifer thickness by analyzing recession hydrographs. Application to the Oman ophiolite Hard-rock aquifer. Journal of Hydrology, 274: 248-269.

Kovacs, A, Perrochet, P, Kiraly, L, Jeannin, P. 2005. A Quantitative Method for Characterization of Karst Aquifers Based on Spring Hydrograph Analysis. J. of Hydrology, 303: 152-164.

In the revised version, we clarified the application of the hydrograph method as follows:

“In the application of the hydrograph method, we looked for straight line segments corresponding to base lows, beginning with the most clear segments (e.g., 4th and 5th segments in Figure 12). These segments allowed us to draw the corresponding fitting lines (dashed red lines). As would be expected, the two lines are parallel because the line slope is solely dependent on the aquifer characteristics and dimension, which are invariant at the timescale of few years (present  case). Based on this result, the fitting lines to the remaining less clear segments were drawn with the same slope and right edge point coincident with the smallest stream flow discharge (Q1). For all years, the point in the graph where the fitting line intercepted the hydrograph at the upper flows (left edge point) was defined as the Q0,t0 point, i.e. the point where the recession period began.”

We also added the suggested references

Discussion/Conclusion

  • English language and style editing may help this section considerably.

The English language was thoroughly revised throughout the entire manuscript.

  • I would love to see the discussion focus on the comparison to the original PUC model as well as to other models that exist.

We added the following paragraph to improve the discussion, namely as regards comparison among model results.

Comparing the first model employed by Costa et al. (2019) with the present application, it is interesting to explain that the inclusion of the water recharge parameter from the PUC method has improved the spacialization of the groundwater recharge, mainly because the first model [10] used total porosity values for each soil class in the Jequitiba River basin [10]. This parameter is not the best to assess the drainage and the water percolation, because according to Silva et al. (2013) the micro and macro porosity reflect better the movement of the water through the soil profile [68]. Moreover, the PUC method includes a management approach which wasn’t taken into consideration in the PF in the first model. Another important difference from the first model to the present one is the separation until the second categorical level of the Neosols class, which has showed a more reliable spacialization and values for the different categorical levels of this soil class. The differences between the physical characteristics (structure, texture and effective depth) of Fluvic Neosols, Litolic Neosols and Quartzarenic Neosols make it questionable to give the same values of hydraulic conductivity and porosity to this soil class.

Reviewer 3 Report

This article presents an application of a spatially explicit model to determine and evaluate the groundwater recharge potential at a catchment scale compatible with the Pandeiros River basin. My comments to the article are as follows: 1. The paper should be carefully revised for punctuation, grammar, spelling mistakes and sentences structuring. 2. Abstract should be more concise. Some general phrases in the abstract about the basic background of article should be placed in the introduction. 3. Authors should include the novel characteristics of the considered fluid model. 4. Add nomenclature before the reference section. 5. Properly number each section in the revised version. 6. Include more explanation for the graphical obtained solutions in the revised draft. 7. Advantages of the applied scheme must be mentioned in the revised version. 8. Similarity index should be reduced in the revised version. Right now, it is almost 30%. Accordingly, I recommend the manuscript is put under minor revision before getting recommended for publication.

Author Response

Reviewer #3

This article presents an application of a spatially explicit model to determine and evaluate the groundwater recharge potential at a catchment scale compatible with the Pandeiros River basin. My comments to the article are as follows:

  1. The paper should be carefully revised for punctuation, grammar, spelling mistakes and sentences structuring.

English language was thoroughly revised throughout the entire manuscript.

  1. Abstract should be more concise. Some general phrases in the abstract about the basic background of article should be placed in the introduction.

The abstract was re-written and is now much more concise.

  1. Authors should include the novel characteristics of the considered fluid model.

We modified a paragraph from the original manuscript to highlight the novelty and motivation of this study. The modified paragraph is reproduced below and clarifies the novelty of this study, relating it with the coupling of physical factors and management practices in a recharge estimation model.

“Despite the positive results obtained by Costa et al. (2019), it is worth noting that the land management practices were a consequence of groundwater recharge assessments, not a contributing factor to groundwater recharge included in the model. Indeed, all parameters included in Costa’s water balance model were physical, while land management practices had no role regardless their potential to dynamically affect groundwater recharge [10]. Thus, the coupling of physical factors and land management practices in a recharge estimation model could be a motivation (and a novelty) for a subsequent study. Before the publication of Costa et al. (2019), Costa et al. (2017) [43] conducted a study in Minas Gerais and developed a method based on multi criteria analysis, which was efficient to map a so-called Conservative Use Potential (PUC). The PUC method weights a considerably large number of variables considering their importance for sustainable land use, including several variables linked to management practices, such as drainage, soil depth and fertility, erosion potential, and land capability. Hence, one possible route to realize our research motivation would encompass including the PUC, as determined by Costa et al. (2017) [43], within the framework of Costa et al. (2019) groundwater recharge method [10].”

  1. Add nomenclature before the reference section.

Nomenclature was added to the revised manuscript before the reference section

  1. Properly number each section in the revised version.

The sections headings were revisited and corrected for numbering when applicable

  1. Include more explanation for the graphical obtained solutions in the revised draft.

The results illustrated in the maps and diagrams were expanded.

  1. Advantages of the applied scheme must be mentioned in the revised version.

The following advantages were added to the conclusions section

“This study can be used as a tool for the management of water resources in the Pandeiros River basin, identifying preferred areas for recharge, thus directing public policies and conservationist actions for these areas. Management practices must be adopted to improve natural groundwater recharge, and hence, increase the available water volume to the local population. The adjustment of irrigation methods, adoption of soil preservation practices to improve water infiltration, seasonal storage of surface water in areas of low recharge potential and the preservation of forest vegetation may include some water management actions. Moreover, this study can provide subsidies for further studies that seek methods for the spatialization of groundwater recharge potential in river basins.”

  1. Similarity index should be reduced in the revised version. Right now, it is almost 30%.

The similarity between stream flow based and PUC based recharges is 84%.

Accordingly, I recommend the manuscript is put under minor revision before getting recommended for publication.

Round 2

Reviewer 1 Report

I would like to recommend the paper for publication without further revisions.

Author Response

Reviewer comment

I would like to recommend the paper for publication without further revisions.

Answer

We are glad that the reviewer was satisfied with the revisions we made. They were very pertinent and helped to improve the original state that is now. Many thanks.

Reviewer 2 Report

Thank you for the opportunity to review the manuscript "An improved model for the evaluation of groundwater recharge based on the concept of Conservative Use Potential: A study in the River Pandeiros watershed, Minas Gerais, Brazil." This manuscript has improved greatly from the previous version and provides interesting insight into modeling groundwater recharge for managers. 

There remain a few issues that I have noted below that could use revision prior to publication.

Introduction

line 129-138: The purpose of the study appears to presume a conclusion. These statements seem more like conclusions, that also require additional support for the assumption. In particular, there needs to be more support for the replacement of coil porosity with texture

Table 3/ Figure 3 - the names used for each unit need to be consistent between these two

What is Cerrado Vegetation? this is an unfamiliar term, defining it would help the reader understand the significance of it more clearly

Methods

The authors describe a step by step process, it would be valuable to include a figure that allows the reader to clearly see each step. I think figure 6 does do this but is not explicitly tied to the methods in the text.

Results

Table 8 - % difference value needs a (.). These results are still questionable - is a 16% difference statistically significant? was this only calculated for one year? Why not use multiple years to more clearly identify similarity? More than a percent difference should be shown if you are claiming similarity between methods - at least a standard deviation of each value (and sample size) is needed.

Figure 12 (and associated text) the red dashed lines do not appear to match the baseflow recession of the system, why are the authors not applying this to a single recession slope (i.e. the flattest slope)? As shown, this does not appear to match with typical applications of the recession slope analysis. Can the authors explain why they did it the way they did as opposed to the standard method?

In general, validation of the models requires a little more rigor - more clearly quantifying the fit of the model to the comparison data set will go a long way to build the argument for the discussion.

Discussion

I like the addition of more comparative analysis with previous models, however this section needs more language and grammar editing to ensure the authors points are clearly shown to the reader. 

Could the authors include a chart that shows similarities and differences and/or pros/cons of each model to help the reader more clearly understand the added value of their modifications?

Conclusion

Instead of just speaking in generalities, the statements of value for water resource management would be greatly supported by clearly articulating what this model had show for management need in the Pandeiros River basin. These specifics would really help the reader understand the general statements more clearly.

Author Response

Reviewer #2

 Comment 1:

Thank you for the opportunity to review the manuscript "An improved model for the evaluation of groundwater recharge based on the concept of Conservative Use Potential: A study in the River Pandeiros watershed, Minas Gerais, Brazil." This manuscript has improved greatly from the previous version and provides interesting insight into modeling groundwater recharge for managers.

Answer

Many thanks for your thorough review. The former comments and suggestions were very pertinent and helped us to improve the manuscript to a much better level of quality. We will carefully address the further comments now.

There remain a few issues that I have noted below that could use revision prior to publication.

Introduction

Comment 2:

line 129-138: The purpose of the study appears to presume a conclusion. These statements seem more like conclusions, that also require additional support for the assumption. In particular, there needs to be more support for the replacement of coil porosity with texture

Answer:

We agree. To improve the revised text we rephrased the following sentence:

“The advantage of the model is that it will presumably more effectively respond to land use changes than the original model.”

That in the revised manuscript is written as

“The replacement has the specific purpose to check whether this set of variables responds more effectively to land use changes than the original variable (porosity).”

Comment 3:

Table 3/ Figure 3 - the names used for each unit need to be consistent between these two

Answer:

Many thanks for the observation. The term “arenite” was replaced in Table 3 and throughout the text by the term “sandstone” as it appears in Figure 3.

Comment 4:

What is Cerrado Vegetation? this is an unfamiliar term, defining it would help the reader understand the significance of it more clearly.

Answer:

We rephrased the sentence where Cerrado Vegetation appears for the first time, and added a reference to the revised text. Where it was

“The spatial distribution of the land use and cover classes in the basin (Figure 5) shows the predominance of typical Cerrado vegetation of low to medium size”

Now is:

“The spatial distribution of the land use and cover classes in the basin (Figure 5) shows the predominance of typical Cerrado vegetation (savanna) of low to medium size [54]”

Methods

Comment 5:

The authors describe a step by step process, it would be valuable to include a figure that allows the reader to clearly see each step. I think figure 6 does do this but is not explicitly tied to the methods in the text.

Answer:

Many thanks for this remark. Figure 6 was improved as to include indication of Steps 1 to 5 in the proper places within the flowchart. We believe now there is a direct link between the steps described in the text and the sectors where they are represented in the figure.

Results

Comment 6:

Table 8 - % difference value needs a (.). These results are still questionable - is a 16% difference statistically significant? was this only calculated for one year? Why not use multiple years to more clearly identify similarity? More than a percent difference should be shown if you are claiming similarity between methods - at least a standard deviation of each value (and sample size) is needed.

Figure 12 (and associated text) the red dashed lines do not appear to match the baseflow recession of the system, why are the authors not applying this to a single recession slope (i.e. the flattest slope)? As shown, this does not appear to match with typical applications of the recession slope analysis. Can the authors explain why they did it the way they did as opposed to the standard method?

In general, validation of the models requires a little more rigor - more clearly quantifying the fit of the model to the comparison data set will go a long way to build the argument for the discussion.

Answer:

Many thanks to this pertinent comment. The application of the hydrograph method was revisited and now applied in a more conventional way, where the lines used to estimate the recession constant and the initial discharge are fitted to the hydrograph baseflow being parallel across the hydrologic years. A note was also made about the calculation of a in the methods section, as usually this parameter is calculated on the timeframe of a log cycle of recharge. The note was this:

“The constant of recession (α; Equation 5) is dependent on the aquifer characteristics and therefore should not vary significantly from year to year. On a Q versus t plot (hydrograph) where the Q values are represented in logarithmic scale and the values in linear scale, the baseflow within a hydrologic year (from the recharge period to the end of recession) should define a straight line the slope of which is related to a. If tcycle is the time of a log cycle for discharge, meaning the time for discharge to change from 1 to 10 m3/s, from 10 to 100 m3/s, and so forth, then  and a = 1/(0.4343tcycle). This simplified representation of Equation 5 is frequently used in the calculation of a and will be adopted in the present study. Contrarily, the values of Q0 can vary in response to the annual variations of precipitation. In this case, a value of Q0 should be calculated for each hydrologic year, while mean ± standard deviation values are derived therefrom.”

Figure 12 (the hydrograph) was also updated, in keeping with your comment. Besides, we added/rephrased a note to explain how the method was applied and recharge calculated.

“In the assessment of recharge using the method of hydrograph recession analysis, we looked for straight line segments corresponding to base lows. These segments allowed us to draw the corresponding fitting lines (dashed red lines). As would be expected, the lines are all virtually parallel because the line slope is solely dependent on the aquifer characteristics and dimension, which are invariant at the timescale of few years (present case). For all years, the point in the graph where the fitting line intercepted the hydrograph at the upper flows (left edge point) was defined as the Q0,t0 point, i.e. the point where the recession period began. On average Q0 = 10.8±1.8 m3 s–1. The coefficient of recession was estimated by the simplified version of Equation 5, a = 1/(0.4343tcycle). In the Pandeiros River basin, the estimated tcycle is 789 days, and therefore a = 0,002918. Using Q0 and a in Equation 6 results for recharge the value V =98.8±20.3 mm year–1.”

The new results are more closer to the spatially distributed recharges, because the difference now is just 5.2%. Thanks again for the comment.

Discussion

Comment 6

I like the addition of more comparative analysis with previous models, however this section needs more language and grammar editing to ensure the authors points are clearly shown to the reader. Could the authors include a chart that shows similarities and differences and/or pros/cons of each model to help the reader more clearly understand the added value of their modifications?

Answer:

The language and grammar was improved throughout the entire manuscript, and especially in this section. We hope this language improvement helps the reader to understand the fundamental differences of using management-based methods of recharge evaluation instead of purely physical methods.

Conclusion

Comment 7

Instead of just speaking in generalities, the statements of value for water resource management would be greatly supported by clearly articulating what this model had show for management need in the Pandeiros River basin. These specifics would really help the reader understand the general statements more clearly.

Answer:

Conclusions were improved. Thanks for the comment